# Dysregulation of Redox Status in Urinary Bladder Cancer Patients

**DOI:** 10.3390/cancers12051296

**Published:** 2020-05-21

**Authors:** Edyta Reszka, Monika Lesicka, Edyta Wieczorek, Ewa Jabłońska, Beata Janasik, Maciej Stępnik, Tomasz Konecki, Zbigniew Jabłonowski

**Affiliations:** 1Department of Molecular Genetics and Epigenetics, Nofer Institute of Occupational Medicine, 91-348 Lodz, Poland; monika.lesicka@imp.lodz.pl (M.L.); edyta.wieczorek@imp.lodz.pl (E.W.); ewa.jablonska@imp.lodz.pl (E.J.); 2Department of Biological Monitoring, Nofer Institute of Occupational Medicine, 91-348 Lodz, Poland; beata.janasik@imp.lodz.pl; 3Department of Toxicology and Carcinogenesis, Nofer Institute of Occupational Medicine, 91-348 Lodz, Poland; maciej.stepnik@imp.lodz.pl; 4Ist Urology Clinic, Medical University of Lodz, 90-549 Lodz, Poland; tomasz.konecki@umed.lodz.pl (T.K.); zbigniew.jablonowski@umed.lodz.pl (Z.J.)

**Keywords:** DNA damage, comet assay, selenium, selenoproteins, NRF2-target genes, DNA methylation and histone deacetylation, gene expression, urinary bladder cancer, blood, recurrence

## Abstract

The alteration of redox homeostasis constitutes an important etiological feature of common human malignancies. We investigated DNA damage, selenium (Se) levels and the expression of cytoprotective genes involved in (1) the KEAP1/NRF2/ARE pathway, (2) selenoprotein synthesis, and (3) DNA methylation and histone deacetylation as putative key players in redox status dysregulation in the blood of urinary bladder cancer (UBC) patients. The study involved 122 patients and 115 control individuals. The majority of patients presented Ta and T1 stages. UBC recurrence occurred within 0.13 to 29.02 months. DNA damage and oxidative DNA damage were significantly higher in the patients compared to the controls, while plasma Se levels were significantly reduced in the cases compared to the controls. Of the 25 investigated genes, elevated expression in the peripheral blood leukocytes in patients was observed for *NRF2*, *GCLC*, *MMP9* and *SEP15*, while down-regulation was found for *KEAP1*, *GSR*, *HMOX1*, *NQO1*, *OGG1*, *SEPW1*, *DNMT1*, *DNMT3A* and *SIRT1*. After Bonferroni correction, an association was found with *KEAP1*, *OGG1*, *SEPW1* and *DNMT1*. Early recurrence was associated with the down-regulation of *PRDX1* and *SRXN1* at the time of diagnosis. Peripheral redox status is significantly dysregulated in the blood of UBC patients. DNA strand breaks and *PRDX1* and *SRXN1* expression may provide significant predictors of UBC recurrence.

## 1. Introduction

Cancer incidence and cancer deaths comprise one of the major public health problems in the world. Urinary bladder cancer (UBC) incidence, diagnosed in both sexes combined, ranks 10th place with 549,393 estimated new cases in 2018. UBC more often affects populations living in developed countries compared to the rest of the world. UBC is more common in men than in women and ranks fourth place among cancers in men in Poland, with 8197 cases (8.6% of new cases in 2018) (Global Cancer Observatory, gco.iarc.fr; [1]). The majority (90–95%) of UBCs—transitional cell carcinomas—comprise superficial tumors (70%), which are usually low-grade and non-invasive (non-muscle invasive bladder cancer, NMIBC) with stage Ta/T1, and the second type has a form of muscle-invasive disease (MIBC), with a stage from T2 to T4 (30%). The overall rate of recurrence for NMIBC is 60% to 70% and the overall rate of progression to a higher stage or grade, and metastasis is 20% to 30% [2].

Oxidative stress with reactive oxygen species (ROS) and also reactive nitrogen species (RNS) generation, and its further consequences, constitutes an important feature of cancer, including UBC etiology, progression, recurrence and metastasis [3]. Redox alteration is an important feature of various cell signaling pathways that are involved in the regulation of cell growth, metabolism, immune regulation and a variety of other physiological functions [4]. One of the recognized redox balance-maintaining pathways is the Kelch-like ECH associated protein 1 (KEAP1)/nuclear factor erythroid 2-related factor 2 (NRF2)/antioxidant response element (ARE) signaling pathway (KEAP1-NRF2-ARE) [5,6]. Under basal conditions, the transcription factor NRF2 is located in the cytoplasm and regulated by inhibitor KEAP1. In response to oxidative stress, NRF2 translocates to the nucleus and, as a consequence, leads to the upregulation of multiple genes with the ARE sequence in their promoters. Hence, the cellular redox homeostasis may be restored [5,6].

The KEAP1/NRF2/ARE regulatory network includes a complex of transcriptional, epigenetic and post-transcriptional levels that turn on during redox perturbation, inflammation, growth factor stimulation and energy fluxes [5,6]. Therefore, it is pointed out that the modulation of the KEAP1/NRF2/ARE pathway may modulate the risk of lifestyle diseases, such as cancer. UBC is predominantly caused by chemical carcinogens derived from cigarette smoking and various occupational and environmental exposures such as to aromatic amines [7]. Chemically induced UBC in rodents has shown the importance of the NRF2 transcription factor in tumor development [8,9,10]. Since it has been widely known that bladder epithelial cells produce many active metabolites, hence the increased expression of *NRF2* was associated with a reduced risk of tumor formation—e.g., UBC. The up-regulation of *NRF2* expression has been also observed in UBC in humans [11].

Selenoproteins may perform an important activity in the prevention of redox balance dysregulation in UBC. Oxidative stress and redox balance regulation are quoted as one of the proposed mechanisms of the anticarcinogenic activity of selenoproteins encoded by selenium (Se) in the form of the 21st amino acid, selenocysteine [12]. Only in limited epidemiological studies has the Se level measured in plasma or serum been indicated as a UBC risk factor. It has been found that the reduction of UBC development risk due to a high Se level, compared to a low level, had the highest value (33% lower risk, 95% CI: 3% to 45%) in comparison to that of other types of cancer [13], but recent findings have indicated that those studies had major weaknesses due to their design, exposure misclassification and potential lifestyle and diet-association unmeasured confounding factors [14]. The role of selenoproteins in UBC is a very rarely investigated issue, and the significance of selenoproteins in UBC is not well recognized yet [15].

DNA damage and oxidative DNA damage is frequently observed in various pathologies, including cancer [16,17]. Data regarding DNA strand breaks in UBC are sparse but profoundly indicate that DNA damage in lymphocytes is elevated in UBC patients [18,19,20]. Additionally, epigenetic mechanisms are clearly involved in the oxidative stress responses in cancer [21].

To sum up, the role of redox status indicators in human UBC etiology and development is still ambiguous, and it requires further research. Thus, we conducted an association study of UBC cases and controls to reveal peripheral redox status alterations in UBC etiology. We focused on the effects of oxidative stress formation in the blood of UBC patients, measured by molecular markers in peripheral blood leukocytes, genomic instability via a comet assay in peripheral lymphocytes and Se levels in the plasma. We examined selected and expected genes involved in (1) the KEAP1/NRF2/ARE pathway, (2) selenoprotein synthesis, and (3) DNA methylation and histone deacetylation (Appendix A), together with DNA damage, oxidative DNA damage and Se levels. Those putative molecular biomarkers of redox status dysregulation were measured and analyzed at the point of diagnosis and analyzed also after the follow-up period of the patients’ surveillance.

## 2. Results

### 2.1. Demographic and Clinicopathological Features

The groups differed in age: 62.88 ± 10.07 years for the UBC patients and 66.38 ± 4.73 years for the controls; *p* = 0.0008; smoking habit: 42 patients currently smoking and 85 alcohol drinkers in the UBC group, and in the reference group, 15 smokers and 100 alcohol drinkers (*p* = 0.0001 and *p* = 0.001, respectively). The groups did not differ in terms of gender distribution and BMI. The mean recurrence was 10.01 ± 7.96 with a minimum of 0.13 and maximum of 29.02 months. Early recurrence occurred in 38 patients (≤1 year after the diagnosis and/or transurethral resection of bladder tumor (TURBT)), and late recurrence was observed in 39 patients, while 22 patients were free from disease recurrence. In 61 patients, Ta stage was observed, in 21, T1 stage, and in 14, T2 stage, while 57 cases had G1 grade, 26, G2 grade and 14, G3 grade. In 72 patients, small (≤3 cm) tumors, and in 40, large (>3 cm) tumors were found, while 71 patients presented a single tumor in the urinary bladder and 38 had multiple tumors (Table 1).

### 2.2. Cytoprotective Gene Expression Analyzed Separately in the UBC Patients and Control Individuals

In peripheral blood leukocytes, the highest relative expression was observed for *GPX1* and *GSTP1*, as well as for *SOD2*, and the lowest, for *OGG1*, *TRXR1* and *UGT1A6*. 

Significant relationships (the most often positive correlations) between the expression of several tested genes in peripheral blood leukocytes and the age, and the BMI index of the examined individuals from both groups were found. Relationships were prevalently observed in the control group.

Higher expression of two genes—*SEP15, SEPW1*—in the women compared to the men was found in both the reference and UBC groups. The control women presented higher expression of six genes compared to the men, while four genes were significantly lower in the women (Appendix A). Current smoking was related to significantly reduced *GPX1* expression in the UBC patients and the controls. The smokers had down-regulation of *OGG1, SOD1, UGT1A6* and *SEPW1* expression when compared to the non-smokers in the control group, while down-regulation in the smokers vs. the non-smokers was observed in the UBC group for *HMOX1* and *SEP15* (Appendix A).

Basic associations between *NRF2* expression and expression of the majority of NRF2-related genes were observed in the UBC and control groups, regardless of cancer disease (Appendix A). 

An association between the Se level and cytoprotective gene expression was observed for *SEPW1* and *KEAP1* in the UBC patients and for *SEP15, GSTP1* and *HMOX1* in the controls.

### 2.3. Alterations in Blood Redox Status of the UBC Patients and Control Individuals

The percentage of DNA damage and oxidative DNA damage was higher in blood of the patients with UBC compared to in the control group (10.81% vs. 7.195% and 6.902% vs. 4.694%, *p* < 0.002 and *p* < 0.002, respectively). The plasma Se concentration was lower in the subjects with UBC compared to in the reference group (67. 52 ± 16.43 vs 74.67 ± 16.98 µg/L, *p* = 0.004) (Table 2)**.**

Alterations of the expression of 13 out of 25 cytoprotective genes in the blood of the UBC patients were observed compared to the reference group. Down-regulation was observable for *KEAP1* (*p* =0.000), *GSR* (*p* = 0.009), *HMOX1* (*p* = 0.003), *OGG1* (*p* = 0.000), *NQO1* (*p* = 0.023), *SEPW1* (*p* < 0.002), *DNMT1* (*p* =0.000), *DNMT3A* (*p* =0.009) and *SIRT1* (*p* = 0.050). By contrast, an increase in expression was characteristic for *NRF2* (*p* = 0.044), *GCLC* (*p* = 0.027), *MMP9* (*p* = 0.021) and *SEP15* (*p* = 0.047). After Bonferroni correction, a significant down-regulation of *KEAP1*, *SEPW1*, *OGG1* and *DNMT1* in UBC remained (Table 2).

The multivariate analysis for Se, DNA and oxidative DNA damage, and all genes presenting statistical differences before the Bonferroni correction revealed a significantly elevated risk of UBC associated with higher DNA damage (OR = 4.96, 95% CI: 2.55–9.62) and oxidative DNA damage (OR = 11.72, 95% CI: 5.16–26.61). A decreased Se level (OR = 6.19, 95% CI: 3.04–12.59) and down-regulation of *KEAP1, OGG1* and *SEPW1* were significantly associated with an increased risk of UBC (OR = 9.83, 95% CI: 4.41–21.91; OR = 3.33, 95% CI: 1.79–6.23; and OR = 6.19, 95% CI: 3.04–12.59, respectively) (Table 3).

### 2.4. The UBC Patients, Clinicopathological Features and Recurrence

We did not observe an association between T stage and G grade and Se level, DNA damage or gene expression in the UBC group. Several significant associations were observed for recurrence time and T stage (Figure 1A). We observed a borderline significant link between G grade and recurrence (*p* = 0.07) with a higher probability of recurrence for the G3 stage. Higher DNA damage was associated with a higher probability of UBC recurrence (*p* = 0.05) (Figure 1B). Lower expression of *SRXN1* (*p* = 0.01) (Figure 1C) and *PRDX1* (*p* = 0.06) (Figure 1D) was associated with a higher probability of UBC recurrence. 

Significant associations of expression and DNA damage and oxidative DNA damage were also found when UBC was divided into early (*N* = 22) and late (*N* = 38) recurrence. Higher expression of *PRDX1* was associated with recurrence more than 1 year after the diagnosis, while *KEAP1* down-expression was associated with early recurrence. Higher DNA damage and oxidative DNA damage were associated with early rather than late recurrence. Additionally, *SEP15* expression was significantly higher in UBC with recurrence (*N* = 60) compared to in the patients without recurrence (*N* = 40).

### 2.5. DNA Damage and Gene Expression in the UBC Patients and Control Individuals

Several associations between DNA damage (Appendix A) or oxidative DNA damage (Appendix A) and cytoprotective gene expression were found in the UBC patients and controls. A positive relationship between *SOD1* expression and tail DNA damage was observed in both groups. Interestingly, *MAFG* and *KEAP1* expression revealed an opposite association in the UBC patients and controls. *MAFG* was positively associated with DNA damage, while *KEAP1* presented a negative association in UBC. In the controls, an opposite pattern was observed; *MAFG* had a negative association, and KEAP1, positive. Interestingly, there were more associations of cytoprotective gene expression with DNA damage than with oxidative DNA damage. Additionally, *PRDX1* and *GCLM* were positively associated with DNA damage and oxidative DNA damage in the UBC group, while *MAFG* was negatively linked to DNA and oxidative DNA damage.

A two-way ANOVA for the further analysis of a potential association between DNA damage, gene expression and UBC showed several significant associations including *MAFG* and *KEAP1* and DNA damage, and oxidative DNA damage. The highest tertile of DNA damage was associated with increased MAFG expression in the UBC patients compared to in the controls, while *KEAP1* expression was reduced in the UBC individuals within the third tertile of DNA damage compared to in the controls presenting the highest DNA damage in the blood (Figure 2).

## 3. Discussion

Oxidative stress, apart from chronic inflammation, constitutes an important mechanism for majority of the common human pathologies, including cancer [4]. In this study, we investigated alterations of redox status in the peripheral blood white cells of the UBC patients compared with that in the control individuals. Furthermore, we explored the impact of the investigated biomarkers at the time of diagnosis on the probability of UBC recurrence. We focused on the investigation of DNA strand breaks and oxidatively damaged DNA in blood lymphocytes, examined by a single cell gel electrophoresis assay (comet assay) and Se levels in the plasma using the IC-PMS assay. Constitutive gene expression was determined in the circulating blood leukocytes of the UBC patients and control individuals using a qRT-PCR assay. We selected genes expected to be involved in (1) the KEAP1/NRF2/ARE pathway, (2) selenoprotein synthesis, and (3) DNA methylation and histone deacetylation (Appendix A).

Cytoprotective genes with the ARE in their promoter region and encoding, e.g., antioxidant, detoxification and repairing enzymes were selected according to the findings from a mouse embryonic fibroblast study that identified a large number of putative NRF2-dependent transcripts involved in various molecular and biological pathways [22]. In the expression analysis, in addition to 14 NRF2-related genes, we also included the *NRF2*, *KEAP1* (NRF2 repressor in cytoplasm) and *MAFG* (NRF2 binding partner that forms heterodimer) genes. Data from the Mammalian Gene Collection Project Team clearly show that the highest mRNA expression in the human urinary epithelium was found for five selenoproteins—*TRXR1*, *GPX1*, *SEP15*, *SELT* and *SEPW1*—in comparison to the rest of the 20 human selenoproteins [23]. Interestingly, it has been revealed that the ARE sequence is present in the promoter region of two selenoproteins: *GPX2* [24] and *TRXR1* [25]. As the representatives of genes involved in epigenetic processes, we examined *DNMT1, DNMT3A*, and *SIRT1* transcript levels.

Oxidative stress is considered the main endogenous source of DNA damage and DNA oxidative damage. Our findings indicated significantly elevated DNA damage and oxidative DNA damage in the lymphocytes of the UBC patients compared with the controls (6.902% vs. 4.694% and 10.81% vs. 7.195%, respectively) (Table 2). Additionally, only two studies conducted on DNA damage in UBC have also revealed higher genomic instability in UBC patients [19,20]. Interestingly, we also observed a prognostic significance of the DNA damage measured by the comet assay prior to the surveillance for UBC recurrence. The patients with the highest level of tail DNA (%) presented a higher recurrence probability (Figure 1B). Interestingly, we did not observe a significant relationship between recurrence and oxidative DNA damage.

The broad cytoprotective effects of NRF2 mediated due to its direct transcriptional targets include not only antioxidant effects, detoxification effects, and modulation of the inflammatory response but also DNA repair [6]. We found a decreased expression of the DNA repair enzyme of the BER pathway—namely, 8-oxoguanine DNA glycosylase 1 (OGG1), which removes oxidatively damaged guanine (8-oxodG) from DNA—in the leukocytes of the UBC patients compared with those in the controls (Table 2). A decreased expression of *OGG1* is related to tumor growth and progression because DNA-repairing ability can very frequently contribute to genomic instability [26]. Additionally, primary findings of Paz-Elizur et al. have clearly indicated that OGG enzymatic activity was lower in peripheral blood mononuclear cells from non-small cell lung cancer patients than in those from control subjects [27]. It is worth mentioning that the analysis of *OGG1* gene expression has shown a weak correlation between OGG enzymatic activity and mRNA expression, indicating the importance of factors other than mRNA expression [28]. *OGG1* expression in the blood has been investigated in various human studies with various pathologies. Down-regulation has been observed in patients with unipolar and bipolar disorder [29,30], Alzheimer’s’ disease [31] and primary open-angle glaucoma [32] compared to in controls. Moreover, our study revealed that *OGG1* expression was lower in the smokers than in the non-smokers, only in the controls (Appendix A). A similar reduction of expression has been found in both cigarette and waterpipe smokers compared to controls [33]. However, not all exposure can result in decreased DNA repairing capacity. For example, in males involved in handling e-waste, *OGG1* expression was up-regulated [34], just as in sulfur mustard-exposed individuals [35]. A specific postprandial oil-based diet in obese people also resulted in the up-regulation of *OGG1* expression [36].

The plasma Se level was significantly reduced in the cases compared to in the controls (67.52 µg/L vs. 74.67 µg/L) (Table 2) and OR = 6.19, 95% CI: 3.04–12.59 or OR = 0.54, 95% CI: 0.30–0.97 in the lowest or the highest categories of Se exposure compared to in the opposite one, respectively (Table 3). This finding is in agreement with previous association studies, suggesting a protective effect of a high Se level against bladder cancer. A meta-analysis of five existing observational bladder cancer studies including, in total, 279,100 individuals and 965 bladder cancer cases showed an inverse association, with an overall risk of OR = 0.67 (95% CI: 0.46–0.97) [13,14]. Another meta-analysis of seven epidemiological studies with 1910 cases and 17,339 controls/cohort members regarding the association between bladder cancer risk and the Se level in the plasma or serum showed that the risk of bladder cancer was inversely associated with the Se level (OR = 0.61, 95% CI: 0.42–0.87) [37]. As was suggested, these epidemiological studies have several confounders, associated with the study design, exposure and lifestyle factors and dietary misclassification [14]. Presumably, the presented association study may carry several confounders because of significant differences in the age (the control group was older than the UBC group) and smoking and alcohol drinking habits (in the control group, smokers and drinkers were less prevalent than in the UBC group). However, all potential confounders were considered in the statistical testing.

It is postulated that the selenoproteins expressed in the urinary epithelium may be involved in UBC etiology and the course of the disease [15]. Additionally, the KEAP1/NRF2/ARE signaling pathway may play the key role in UBC development, progression and recurrence [5,6]. Animal studies have shown that NRF2 plays a central role in cancer prevention due to the modulation of susceptibility to chemical carcinogenesis in carcinogen-exposed NRF2-null mice [8,10,38]. We observed the altered expression of 13 of the 25 investigated genes in peripheral blood leukocytes in the UBC patients. Elevated expression was observed for *NRF2*, *GCLC*, *MMP9* and *SEP15*, while down-regulation was found for *KEAP1*, *GSR*, *HMOX1*, *NQO1*, *OGG1*, *SEPW1*, *DNMT1*, *DNMT3A* and *SIRT1*. However, after Bonferroni correction, associations were found only between the NRF2 repressor *KEAP1*, NRF2 target *OGG1*, selenoprotein *SEPW1*, and a gene involved in DNA methylation, *DNMT1* (Table 2 and Table 3).

Alterations of NRF2 and other KEAP1/NRF2/ARE pathway-related genes in UBC may indicate a redox imbalance due to oxidative stress formation. Previous studies have also indicated that NRF2-modulated gene expression significantly differed in the peripheral blood of UBC patients compared with in controls [39]. Although NRF2 deficiency led to carcinogenesis, the increased expression of NRF2 can lead to resistance to chemotherapy and may predict a poor prognosis [40,41,42]. Recent findings have indicated that NRF2 and its negative regulator KEAP1 are frequently mutated in cancer [43], and mutations have been also observed in UBC [44]. Therefore, the constitutive activation of the KEAP1/NRF2/ARE pathway may confer chemo- and radio-resistance in the treated patients. That dual role of NRF2 at different stages of cancer disease may reflect the flexible redox-maintaining mechanisms. A similar effect has been proposed for Se and also selenoproteins. Both an excess of and deficiency in Se may be reflected as a U curve (hormetic curve) and may influence the risk of cancer [45,46]. We observed that *GPX1, GPX3, SEP15* and *SEPP1* selenoprotein gene expression in the leukocytes of the UBC males was significantly lower compared to in the healthy control males, which may suggest the alteration of Se status in the blood of patients. Similarly, a significantly lower plasma Se level in the investigated 33 UBC males than in the control males was observed [47]. In the present study, among the five investigated selenoprotein encoding genes, only *SEPW1* expression was significantly down-regulated in the peripheral blood leukocytes of the UBC patients when comparing to the controls. Recently, increasing attention has been paid to *SEPW1*, which is a highly conserved protein with a thioredoxin-like fold and a CXXU motif, among eight known selenoproteins, including *SELH, SELM, SELO, SEPP1, SELT, SELV, SEPW1* and *SEP15*. This specific structure is responsible for maintaining the redox homeostasis of a cell, or it participates in protein folding [48]. SEPW1 is required for normal cell cycle progression and presents antioxidative activity because it can reduce hydrogen peroxide by reduced glutathione [49]. A cytosolic selenoprotein, SEPW1, is ubiquitously expressed, but it has been primarily found in the muscles, i.e., the muscles surrounding the urinary bladder [23]. SEPW1 deficiency is able to reduce ubiquitination or facilitate the phosphorylation of TP53, resulting in an increased level of activated TP53 and G1 cell cycle arrest [50,51]. Recent studies have reported that *SEPW1* expression is reduced in human cancers: breast, hepatocellular and gastric [52,53,54,55]. Moreover, the down-regulation of *SEPW1* has been associated with the poor survival of patients with lung adenocarcinoma [56]. Although *SEPW1* and *GPX1* are very sensitive to a low Se supply [46,48], we did not observe an association between *SEPW1* expression and the plasma Se level. In this context, the depressed *SELW1* expression observed in the blood of the UBC patients may function in the regulation of the UBC-related redox balance in a manner not related to the Se level. In addition, although the Se status and KEAP1/NRE2/ARE pathway can interplay, especially in the depletion of Se [57], we did not observe an association between Se and investigated cytoprotective genes, probably because of the physiological plasma Se level presented by the study volunteers.

Previous studies have indicated the overexpression of *DNMT1* in UBC, when measured in urinary bladder tissues [58,59]. Moreover, DNMT1 protein expression has been proposed to be a significant clinical predictor for the stage and treatment response of UBC [59]. Therefore, our findings of a significant down-regulation of *DNMT1* in leukocytes may point out alterations in peripheral epigenetic regulation. Peripheral blood leukocytes constitute an easily assessable biological material and can be regarded as a substitute target organ of action, as has been demonstrated in human studies. Many studies have examined the ability of gene expression levels measured in circulating leukocytes to predict, diagnose and stratify disease outcomes [60,61]. It should be noted that each leukocyte subset presents a unique pattern of gene expression related to its specific function. For gene expression assays, we used the total RNA from white blood cell populations (PAXGene™ Blood RNA tubes), including lymphocytes, monocytes and granulocytes. Thus, the different expression of cytoprotective genes in the UBC patients and controls reported here may reflect the relative proportions of specific leukocyte subpopulations.

In addition to high DNA damage in peripheral lymphocytes (Figure 1B), the probability of early UBC recurrence was associated with the down-regulation of the expression of NRF2 targets—*PRDX1* (Figure 1D) and *SRXN1* (Figure 1C)—at the time of diagnosis. PRDX1 is a member of an ubiquitous family of six peroxiredoxins, a family of thiol-based peroxidases, redox-regulating proteins involved in defense against oxidative stress. Similarly to SEPW1, PRDX1 is engaged in regulating cell growth, differentiation and apoptosis. Recent studies have found that PRDX1-regulated pathways play an important role in the progression and metastasis of various cancer types [62]. The up-regulation of *PRDX1* has been observed in bladder cancer, breast cancer, colorectal cancer, lung cancer, gastric cancer, liver cancer, pancreatic cancer, sarcoma, leukemia and lymphoma, while down-regulation was detected in esophageal, head and neck cancers and myeloma [63]. The sulfiredoxin SRXN1 is a small redox protein involved in the redox-activated thiol switch in certain peroxiredoxins (2-Cys PRX), namely PRDX1-4. It provides a reduction reaction for cysteine (Cys) residues after reversible oxidation [64]. Moreover, SRXN1 catalyzes the deglutathionylation of several distinct proteins in response to oxidative stress [65]. SRXN1 is also involved in cancer in a peroxiredoxin-dependent and independent manner [66,67]. The over-expression of *SRXN1* has been found in various cancers, including breast, colorectal, lung, prostate and skin cancers, and down-regulation has been found in esophageal cancer [63].

Interestingly, as proposed by Mishra et al., both peroxiredoxins and sulfiredoxin may interact and collaborate to maintain cellular redox balance and modulate cell signal transduction and cancer development. They concluded that the expression levels of individual components of the SRXN–PRDX axis have been correlated with patient survival outcomes in multiple cancer types [63]. Similarly, our results regarding *PRDX1* and *SRXN1* expression in the peripheral blood of the UBC patients indicate the significance of peroxiredoxin and sulfiredoxin for UBC recurrence. It seems important to point out novel biological mechanisms in the etiology of UBC and to develop molecular biomarkers for early UBC detection, prognosis and effective treatment. Thus, further investigation of SRXN–PRDX molecular cross-talk may be of interest to study the significance of this thiol-based system in UBC.

## 4. Materials and Methods

### 4.1. Study Design

The study involved 122 patients with urinary bladder cancer (UBC) over 60 years of age, admitted to the Ist Department of Urology at the Medical University in Lodz (Hospital of Military Memorial Medical Academy—Central Veterans’ Hospital) in the period from May 2013 to May 2016. One hundred and fifteen individuals from a similar age group, who volunteered to study in the Nofer Institute of Occupational Medicine (NIOM, Lodz), constituted the reference group. Venous blood samples were collected once into BD Vacutainer^®^ tubes (EDTA and PAXgene™ Blood RNA tubes) before transurethral resection of bladder tumor (TURBT), cystectomy or study involvement, respectively. UBC diagnosis was histopathologically confirmed.

During the study, the TURBT or cystectomy patients underwent a medical interview and cystoscopy for diagnosis and clinical parameters (T stage and G grade, number and size of tumors, recurrence). The patients were diagnosed as non-muscle invasive bladder cancer with Ta and T1 stage or muscle invasive bladder cancer with T2 stage. According to the procedure for the treatment of the patients with bladder cancer, cystoscopy in the patients with diagnosed bladder cancer was performed 2 years after the diagnosis—every 3 months, and in the third year of the study follow up—every 4–6 months.

The questionnaire survey, including history of tobacco smoking, occupational exposure, past diseases, dietary habits, drug use etc., was conducted in both groups of the UBC patients and controls. This study was approved by the Bioethics Committee at the Nofer Institute of Occupational Medicine in Lodz (No. 10/2012; date: 6 May 2012).

### 4.2. Comet Assay

DNA damage including strand breaks and alkali-labile sites was assayed in the whole blood using the alkaline single-cell gel electrophoresis (SCGE; comet assay) method as described by Singh et al. [68] and modified by Mc Kelvey-Martin et al. [69]. The level of DNA breaks was analyzed by means of a comet assay (alkaline single cell gel electrophoresis) in peripheral blood lymphocytes. Additionally, an assessment of oxidatively damaged DNA in terms of oxidized purines converted to strand breaks with formamidopyrimidine DNA glycosylase (FPG, New England Biolabs Inc, Ipswich, MA, USA) was performed. The following comet parameters were analyzed: tail moment, tail length, % of DNA in the tail and the numbers of comets (cells with an arbitrary cut-off value of head intensity < 90% of DNA content) by means of an Olympus fluorescence microscope (a BX40 instrument; Olympus, Tokyo, Japan) equipped with an image analysis system (Comet IV, Perceptive Instruments, UK). For each participant, four slides were prepared simultaneously: two for the assessment of DNA strand breakage and the other two, which also included the FPG treatment, for the assessment of total DNA damage (i.e., DNA strand breakage and oxidatively generated DNA damage). Respective DNA damage was inferred based on the relative amount of DNA in the comet tail (henceforth referred to as % DNA) obtained via computer-aided image analysis.

### 4.3. Gene Expression

Total RNA was isolated from venous blood using a PAXgene RNA Blood Mini Kit (PreAnalytiX GmbH, Hombrechtikon, Switzerland). The constitutive mRNA expression of a panel of 25 genes involved in the 3 pathways: (1) the KEAP1/NRF2/ARE pathway, (2) selenoprotein encoding, and (3) DNA methylation and histone deacetylation were determined using quantitative real-time PCR (qRT-PCR). The cDNA from 100 ng of total RNA was synthesized with the Transcriptor First Strand cDNA Synthesis Kit (Roche, Basel, Switzerland) using the PTC-200 thermocycler (MJ Research, Bio-Rad Laboratories, Inc., Berkeley, CA, USA).

All the samples were amplified in duplicate. Randomly selected samples were simultaneously amplified on the same plate to determine the inter-assay and intra-assay coefficients of variability (CV). Expression was quantified with the FastStart Essential SYBR Green Master (Roche, Basel, Switzerland) using the QuantStudio™ 12K Flex Real-Time PCR System (Thermo Fisher Scientific, Waltham, MA, USA), Carlsbad, CA, USA). Among the panel of 6 house-keeping genes: NM_002046.3 glyceraldehyde-3-phosphate dehydrogenase (*GAPDH*), NM_000181.3 glucuronidase, beta (*GUSB*), NM_000291.3 phosphoglycerate kinase 1 (*PGK1*), NM_004048.2 beta-2-microglobulin (*B2M*), NM_012423.2, ribosomal protein L13a (*RPL13A*), NM_003194.4 TATA box binding protein (*TBP*), *GAPDH* and *RPLP0* were selected as reference genes, according to the Ct and CV values. The primers for the target genes were designed with the Beacon Designer 7.0 (PREMIER Biosoft International, San Francisco, CA, USA), and each amplicon < 100 bp sequence would have complied with exon–exon boundaries.

Gene expression data were evaluated by a Livak-based method, the ∆Ct method with reference to gene-normalized relative quantification. The Ln transformation of gene expression values was applied to normalize the distribution. All the gene expression procedures fulfilled the MIQE guidelines (Minimum Information for Publication of Quantitative Real-Time PCR Experiment).

A list of the examined genes with their functions is presented in Appendix A. For NRF2 and selected selenoprotein-encoding genes, we applied the common names. Additionally, *GPX2* and *DNMT3B* expression in the peripheral blood tleukocytes was below the detection limit for the qRT-PCR assay.

### 4.4. Se Measurement

Inductively Coupled Plasma Mass Spectrometry (ICP-MS) applied with ELAN DRC-e with Dynamic Reaction Cell (Perkin Elmer, SCIEX, Waltham, MA, USA) was used for the determination of Se. The blood plasma was prepared by 150-fold dilution in nitric acid (1%). The certificate reference material—human serum BCR-637 from the National Institute of Standard and Technology (NIST)—was examined for Se determination.

### 4.5. Statistical Analysis

Continuous variables were expressed as means with standard deviations (SD). Normality was assessed using the Shapiro–Wilk W-test. Variables that were not normally distributed were transformed by means of Box-Cox transformation.

To compare qualitative variables between the bladder cancer patients and control group, the Pearson chi-square test was used. Differences between the two groups—the bladder cancer patients vs. control group—were evaluated by Student’s t-test or the Mann–Whitney U test. Additionally, the p-value was adjusted for confounding factors including age, gender, BMI, smoking status and alcohol drinking status by linear regression. Separate statistical models for each gene adjusted for the aforementioned variables were used. Significance was declared after Bonferroni correction for 25 genes, *p*-value < 0.002.

Moreover, linear regression was used to evaluate the association between selenium status, gene expression levels and DNA damage. Spearman correlation was used to compare the quantitative data obtained from the gene expression and DNA damage assays. Comparisons between multiple groups were performed using the analysis of variance (ANOVA) or the Kruskal–Wallis one-way analysis of variance. Additionally, the DNA damage and gene expression levels were classified as “low”, “medium” or “high” according to the distribution of the selected cytoprotective gene’s expression level, % tail DNA damage and oxidative % tail DNA damage in the control group. The “low” classification corresponds to the values ≤ the 33rd percentile (Tertile 1); “medium”, to the values between the 33rd percentile and the 66th percentile (Tertile 2); and “high”, to the values > the 66th percentile (Tertile 3) (Appendix A). Moreover, the two-way ANOVA was used to present associations between DNA damage and cytoprotective gene expression. Odds ratios (OR) and 95% confidence intervals (95% CI) were calculated by multivariate logistic regression for redox status indices in the urinary bladder cancer patients, adjusted by age, BMI, and smoking and alcohol habits. Cancer recurrence was analysed by the plotting Kaplan–Meier curves, and the recurrence-free survival probability distributions were compared using the log-rank test. Statistica 12.0 (StatSoft, Tulsa, OK, USA) was used for the statistical analysis.

## 5. Conclusions

The results of this association study clearly indicate the dysregulation of redox status in the blood of the UBC patients. Additionally, apart from alterations in the Se status, DNA strand breaks and oxidative DNA strand breaks, our findings also indicate the significance of specific genes involved in the KEAP1/NRF2/ARE pathway—*KEAP1*, *OGG1*, the selenoprotein encoding gene *SEPW1* and the DNA methylase *DNMT1*—in UBC etiology. Selected NRF2 targets that encode antioxidative proteins—*PRDX1* and *SRXN1*—may play an important role in UBC recurrence. The diminished expression of both genes in the peripheral blood of UBC patients led to early rather than late recurrence. Therefore, this thiol-based SRXN–PRDX system may be involved in the abnormal proliferation of urothelial cells.

The diagnostic treatment of UBC still relies, essentially, on cystoscopy, since no relevant panel of predictive UBC biomarkers is available for tumor detection, the prognosis of recurrence and progression, or the prediction of response to the therapy (e.g., for MIBC). Thus, KEAP1/NRF2/ARE, selenoprotein synthesis and epigenetic pathway activation and their cytoprotective activities may provide promising targets for the prevention as well as the therapy of UBC. Additionally, the diagnostic measurement of the plasma Se level seems to be of value, but the semi-quantitative comet assay may be difficult for clinical implementation. In summary, because of sparse investigations, the role of NRF2 in UBC etiology and UBC recurrence seems to be still ambiguous, and our findings require further research. The reliability of tested redox status biomarkers must be proved in larger UBC patient cohorts in case-control and prospective epidemiological studies.

## Figures and Tables

**Figure 1 cancers-12-01296-f001:**
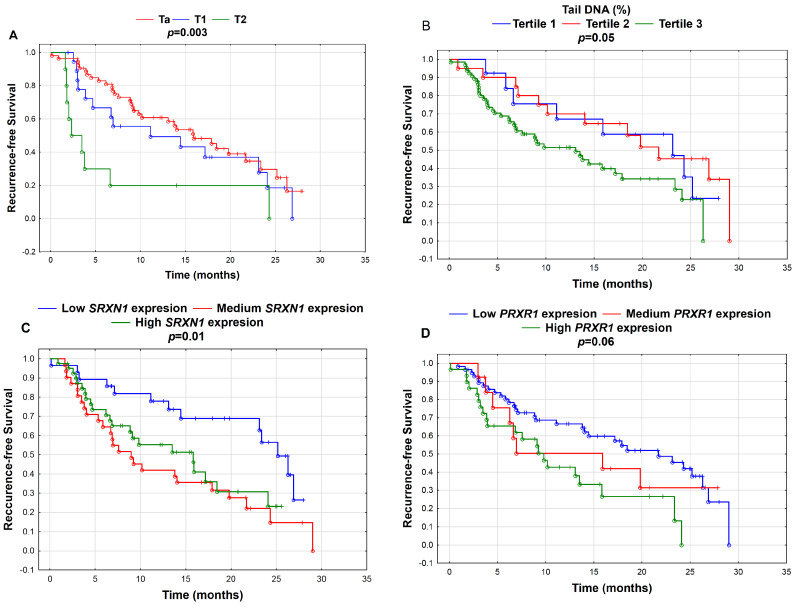
Kaplan–Meier curves for recurrence-free survival according to (**A**) tumor stage (Ta, T1, T2); (**B**) tertiles of tail DNA (%)—Tertile 1 (≤3.74% Tail DNA; *N* = 7), Tertile 2 (3.75–5.11% Tail DNA; *N* = 24) and Tertile 3 (>5.1% Tail DNA; *N* = 91); (**C**) tertiles of gene expression of *SRXN1*—low *SRXN1* expression (≤7.68; *N* = 35), medium *SRXN1* expression (7.68–8.03; *N* = 35) and high *SRXN1* expression (>8.03; *N* = 52); and (**D**) tertiles of gene expression of *PRDX1*—low *PRDX1* expression (≤9.47; *N* = 66), medium *PRDX1* expression (9.47–9.61; *N* = 16) and high *PRDX1* expression (>9.61; *N* = 40). P-values were calculated by the log-rank test; *p* < 0.05.

**Figure 2 cancers-12-01296-f002:**
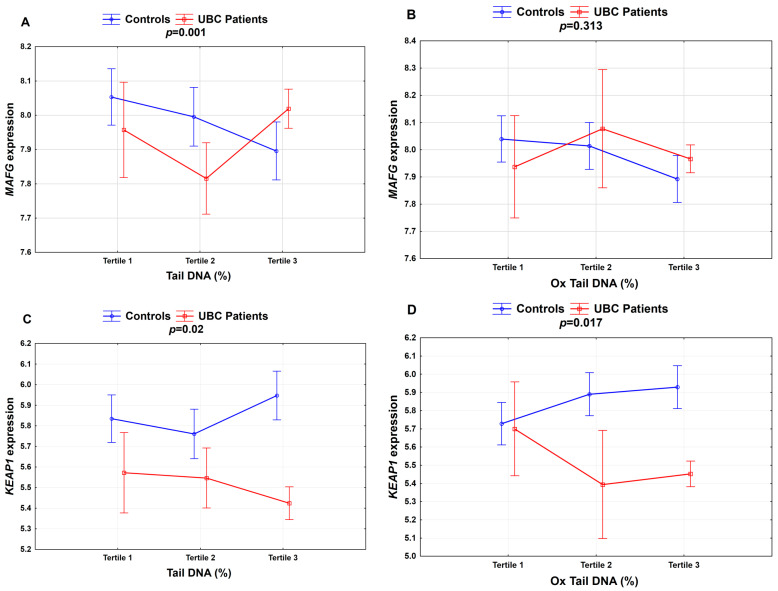
Association between DNA damage and selected cytoprotective gene expression in the study group for (**A**) *MAFG*; (**C**) *KEAP1*—Tertile 1 (≤3.74% Tail DNA; *N* = 7), Tertile 2 (3.75–5.11% Tail DNA; *N* = 24) and Tertile 3 (>5.1% Tail DNA; *N* = 91); (**B**) *MAFG*; and (**D**) *KEAP1*—Tertile 1 (≤6.21% Ox Tail DNA; *N* = 8), Tertile 2 (6.22–7.37% Ox Tail DNA; *N* = 6) and Tertile 3 (>7.37% Ox Tail DNA; *N* = 108). *p*-values were calculated by the two-way ANOVA *p* < 0.05; the percentages of DNA damage (Tail DNA) and oxidative DNA damage (Ox Tail DNA).

**Table 1 cancers-12-01296-t001:** Demographic and histopathological features.

Characteristic	Urinary Bladder Cancer	Control	*p*-Value
	*N* = 122	*N* = 115	
Age mean ± SD	62.88 ± 10.07	66.38 ± 4.73	0.0008 ^1^
Gender			
Males	86 (70.49%)	81 (70.43%)	NS
Females	36 (29.51%)	34 (29.57%)	
BMI mean ± SD	27.29 ± 4.60	26.79 ± 3.25	NS
Smoking habit			
Yes	42 (34.42%)	15 (13.04%)	0.0001 ^2^
No	80 (65.57%)	100 (86.96%)	
Alcohol			
Yes	85 (69.67%)	100 (86.96%)	0.001 ^2^
No	37 (30.33%)	15 (13.04%)	
T stage			
Ta	61 (50.0%)		
T1	21 (17.21%)		
T2	15 (12.30%)		
Unknown	25 (20.49%)		
G grade			
G1	57 (46.72%)		
G2	26 (21.31%)		
G3	14 (11.48%)		
Unknown	25 (20.49%)		
Recurrence			
Recurrence months	10.01 ± 7.96		
Early <1 year	38 (31.15%)		
Late recurrence ≥1 year	39 (31.97%)		
Free	22 (18.03%)		
Unknown	23 (18.85%)		
No of tumors			
Single	71 (58.20%)		
Multiple	38 (31.15%)		
Unknown	13(10.65%)		
Size of tumors			
Small (≤3 cm)	72 (59.01%)		
Large (>3 cm)	40 (32.79%)		
Unknown	10 (8.20%)		

^1^ The *p*-values were calculated using the Student’s *t*-test; ^2^ the *p*-values were calculated using the Pearson Chi-Square test; NS, not significant.

**Table 2 cancers-12-01296-t002:** Redox status in the blood of the Urinary bladder cancer patients and the controls.

Redox Parameter	Urinary Bladder Cancer *N* = 122	Control *N* = 115	Beta (ß) Coefficient	*p*-Value ^1^
Se ^2^	67.52 ± 16.43	74.67 ± 16.98	−0.204	0.004
*NRF2*	8.540 ± 0.337	8.448 ± 0.257	0.142	0.044
*KEAP1*	5.466 ± 0.336	5.848 ± 0.410	−0.460	**0.000**
*MAFG*	7.970 ± 0.281	7.983 ± 0.261	−0.047	0.511
*ABCC4*	5.932 ± 0.744	5.973 ± 0.473	−0.056	0.431
*GCLC*	6.427 ± 0.398	6.307 ± 0.318	0.153	0.027
*GCLM*	7.834 ± 0.346	7.890 ± 0.224	−0.122	0.089
*GSR*	7.189 ± 0.371	7.280 ± 0.524	−0.184	0.009
*GSTP1*	10.390 ± 0.260	10.420 ± 0.200	−0.013	0.851
*HMOX1*	8.649 ± 0.357	8.820 ± 0.256	−0.199	0.003
*MMP9*	8.475 ± 0.824	8.201 ± 0.606	0.164	0.021
*NQO1*	5.461 ± 0.509	5.620 ± 0.486	−0.160	0.023
*OGG1*	3.144 ± 0.479	3.541 ± 0.692	−0.316	**0.000**
*PRDX1*	9.491 ± 0.226	9.534 ± 0.158	−0.070	0.309
*SOD1*	9.075 ± 0.306	9.064 ± 0.239	0.010	0.884
*SOD2*	11.00 ± 0.53	10.91 ± 0.429	0.134	0.060
*SRXN1*	7.944 ± 0.492	7.874 ± 0.534	0.063	0.379
*UGT1A6*	3.393 ± 0.861	3.572 ± 1.051	−0.036	0.620
*GPX1*	11.82 ± 0.55	11.72 ± 0.42	0.092	0.182
*SELT*	7.371 ± 0.248	7.363 ± 0.314	0.091	0.200
*SEP15*	9.108 ± 0.161	9.090 ± 0.142	0.137	0.047
*SEPW1*	6.058 ± 0.335	6.480 ± 0.439	−0.497	**0.000**
*TRXR1*	5.151 ± 0.482	5.170 ± 0.344	−0.097	0.170
*DNMT1*	6.696 ± 0.314	6.869 ± 0.260	−0.320	**0.000**
*DNMT3A*	6.316 ± 0.403	6.408 ± 0.306	−0.187	0.009
*SIRT1*	6.191 ± 0.274	6.262 ± 0.261	−0.139	0.050
Tail DNA (%) ^3^	6.902 ± 2.741	4.694 ± 1.705	0.446	0.000
Ox Tail DNA (%) ^4^	10.810 ± 3.350	7.195 ± 2.001	0.536	0.000

^1^ The *p*-values were calculated by a linear regression on Box-Cox transformed values adjusted by age, gender, BMI, and smoking and alcohol habits; separate statistical models for each gene were used; ^2^ plasma selenium level (µg/L); values in bold are significant after Bonferroni correction for 25 genes, *p*-value < 0.002; *NQO1*—1 missing patient; *UGT1A6*—6 missing patients and 113 controls; *OGG1*—1 missing patient; the percentages of DNA damage ^3^ and oxidative DNA damage ^4^.

**Table 3 cancers-12-01296-t003:** Odds ratio multivariate analyses for redox status indices in the urinary bladder cancer patients.

Variables	Median	Group	≤Median *N*	Median< *N*	Cut-Off Value ^2^	OR	95.00%	95.00%	*p*-Value ^1^
Se ^3^	75.5	Co UBC	58 81	57 41	≤Median	6.19	3.04	12.59	0.04
*NRF2*	8.46	Co UBC	59 48	56 74	>Median	1.57	0.88	2.80	0.13
*KEAP1*	5.89	Co UBC	58 110	57 12	≤Median	9.83	4.41	21.91	**0.0000**
*GCLC*	6.32	Co UBC	58 56	57 66	>Median	1.21	0.68	2.16	0.51
*GSR*	7.31	Co UBC	58 73	57 49	≤Median	1.44	0.81	2.57	0.21
*HMOX1*	8.82	Co UBC	58 86	57 36	≤Median	2.23	1.22	4.07	0.009
*MMP9*	8.24	Co UBC	59 44	56 78	>Median	2.00	1.11	3.63	0.02
*NQO1*	5.64	Co UBC	58 82	57 39	≤Median	2.38	1.31	4.33	0.005
*OGG1*	3.51	Co UBC	58 94	57 27	≤Median	3.33	1.79	6.23	**0.0002**
*SEP15*	9.08	Co UBC	59 51	56 71	>Median	1.83	1.02	3.30	0.04
*SEPW1*	6.52	Co UBC	58 106	57 16	≤Median	6.19	3.04	12.59	**0.0000**
*DNMT1*	6.83	Co UBC	58 80	57 42	≤Median	1.89	1.06	3.37	0.03
*DNMT3A*	6.41	Co UBC	59 75	56 47	≤Median	2.03	1.11	3.69	0.02
*SIRT1*	6.27	Co UBC	58 73	57 49	≤Median	1.56	0.88	2.77	0.13
Tail DNA ^4^	4.26	Co UBC	58 57	22 100	>Median	4.96	2.55	9.62	0.0000
Ox Tail DNA ^5^	6.76	Co UBC	58 10	57 112	>Median	11.72	5.16	26.61	0.0000

^1^ The *P*-values were calculated by a logistic regression adjusted by age, BMI, and smoking and alcohol habits, separately for each gene; ^2^ cut-off median values for the >median and ≤median of controls; UBC—Urinary Bladder Cancer; Co—Control group; ^3^ plasma selenium level (µg/L), OR = 0.54, 95% CI: 0.30–0.97 in the highest category of Se exposure compared to in the lowest one; values in bold are significant after Bonferroni correction for 25 genes, *p*-value < 0.002; the percentages of DNA damage ^4^ and oxidative DNA damage ^5^.

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
