# Peer review of "Dysregulation of Redox Status in Urinary Bladder Cancer Patients"

_cancers, 2020, doi:10.3390/cancers12051296_

Round 1

Reviewer 1 Report

The paper is dedicated to the linkage of redox status in peripheral blood to the UBC. This is an important study because of the necessity of non-invasive analysis of disease progression. The design of the study is adequate because selenium level, DNA damage, and expression of tested genes are together good markers for redox status. However, it is difficult to predict the applicability of the assays to the clinic. The analysis of selenium level is durable, but semi-quantitative comet assay is difficult to adopt for the routine clinical analysis. The reliability of tested marker genes must be proved on higher patient cohorts. The most interesting part of the study: relations of SPXN1 and PRDX1 with tumor reccurance must be better articulated.

Author Response

Dear Reviewer,

We would like to thank you for reviewing our paper and valuable comments and suggestions. We hope that the revised version of the manuscript meets your expectations and that our answers for all comments are satisfying. Changed parts of the manuscript are marked in red font.

Kind regards,

Edyta Reszka

The paper is dedicated to the linkage of redox status in peripheral blood to the UBC. This is an important study because of the necessity of non-invasive analysis of disease progression. The design of the study is adequate because selenium level, DNA damage, and expression of tested genes are together good markers for redox status. However, it is difficult to predict the applicability of the assays to the clinic. The analysis of selenium level is durable, but semi-quantitative comet assay is difficult to adopt for the routine clinical analysis. The reliability of tested marker genes must be proved on higher patient cohorts. The most interesting part of the study: relations of SPXN1 and PRDX1 with tumor reccurance must be better articulated.

Authors’ response: We absolutely agree that at a present moment 1) it is difficult to predict the applicability of the assays to the clinic and 2) relations of SPXN1 and PRDX1 with tumor reccurance must be better articulated. Thus, we have emphasized a little more these two issues in the last paragraph of Discussion (on 6th page) and Conclusions Section (on 8th page).

Reviewer 2 Report

The study is well designed. Statistical analyses are well done and well supports the investigation of initial hypothesis.

Major comments:

M&Ms:

1) In lines 426-427 Authors report: "The patients were diagnosed as non-muscle invasive bladder cancer with low T1 stage and muscle invasive bladder cancer with a stage from T2 to T4".
I do not understand what is low T1. Do You mean low-grade T1? If is it, what about T1 high-grade.
Secondly, do not use the connector "and", but you should use "or", in fact patient can be diagnosed for either T1 or T2-T4

2) Authors report that some statistical models were adjusted for confounding factors including age, gender, BMI, smoking status, alcohol drinking status by linear regression. However, it is not clear if Authors repeated separate models for each gene adjusted for the aforementioned variables, or if all genes were put in the same model. If I'm correct and separate tests were performed, please clarify it in the materials and methods section and in table legends.  The second option cannot be taken into account because of the number of events needed to test all the genes. 

Minor comments:

1) Author should be more precise with references. Particularly intro beginning they leave entire periods (plenty of data) without any ref. Please try to fix it even at the cost of repeating some references. 

2) In lines 167-68 You report: "Higher DNA damage was associated with the higher probability of recurrence of DNA damage", may be you meant "recurrence of cancer" and latter "of DNA damage" is a mistake.

Author Response

Dear Reviewer,

We would like to thank you for reviewing our paper and valuable comments and suggestions. We hope that the revised version of the manuscript meets your expectations and that our answers for all comments are satisfying. Changed parts of the manuscript are marked in red font.

Kind regards,

Edyta Reszka

Major comments:

M&Ms:

1) In lines 426-427 Authors report: "The patients were diagnosed as non-muscle invasive bladder cancer with low T1 stage and muscle invasive bladder cancer with a stage from T2 to T4".
I do not understand what is low T1. Do You mean low-grade T1? If is it, what about T1 high-grade.
Secondly, do not use the connector "and", but you should use "or", in fact patient can be diagnosed for either T1 or T2-T4.

Authors’ response: We have corrected the description of T stage histopathology. Actually, our association study includes Ta, T1 patients (NMIBC) and T2 patients (MIBC) (see corrected sentence on 6th page and Table 1). We are sorry about that mistake.

2) Authors report that some statistical models were adjusted for confounding factors including age, gender, BMI, smoking status, alcohol drinking status by linear regression. However, it is not clear if Authors repeated separate models for each gene adjusted for the aforementioned variables, or if all genes were put in the same model. If I'm correct and separate tests were performed, please clarify it in the materials and methods section and in table legends.  The second option cannot be taken into account because of the number of events needed to test all the genes. 

Authors’ response: Thank you for this valuable comment. We have included information about statistical models that had been applied separately for each gene (see on 7th page and relevant Tables caption).

Minor comments:

  • Author should be more precise with references. Particularly intro beginning they leave entire periods (plenty of data) without any ref. Please try to fix it even at the cost of repeating some references. 

Authors’ response: We have included two additional missing references regarding oxidative stress (H. Sies) and selenium (R. Brigelius-Flohe) and we have refined references on KEAP1/NRF2/ARE pathway.

  • In lines 167-68 You report: "Higher DNA damage was associated with the higher probability of recurrence of DNA damage", may be you meant "recurrence of cancer" and latter "of DNA damage" is a mistake.

Authors’ response: Thank you. This mistake has been corrected.

Round 2

Reviewer 2 Report

All my comments have been finely answered.